# Preparation and Characterization of Thermally Stable Collagens from the Scales of Lizardfish (*Synodus macrops*)

**DOI:** 10.3390/md19110597

**Published:** 2021-10-21

**Authors:** Junde Chen, Guangyu Wang, Yushuang Li

**Affiliations:** Technical Innovation Center for Utilization of Marine Biological Resources, Third Institute of Oceanography, Ministry of Natural Resources, Xiamen 361005, China; 17859733637@163.com (G.W.); liyushuang@tio.org.cn (Y.L.)

**Keywords:** lizardfish scale, marine collagen, thermal stability, cell viability

## Abstract

Marine collagen is gaining vast interest because of its high biocompatibility and lack of religious and social restrictions compared with collagen from terrestrial sources. In this study, lizardfish (*Synodus macrops*) scales were used to isolate acid-soluble collagen (ASC) and pepsin-soluble collagen (PSC). Both ASC and PSC were identified as type I collagen with intact triple-helix structures by sodium dodecyl sulfate-polyacrylamide gel electrophoresis and spectroscopy. The ASC and PSC had high amino acids of 237 residues/1000 residues and 236 residues/1000 residues, respectively. Thus, the maximum transition temperature (T_max_) of ASC (43.2 °C) was higher than that of PSC (42.5 °C). Interestingly, the T_max_ of both ASC and PSC was higher than that of rat tail collagen (39.4 °C) and calf skin collagen (35.0 °C), the terrestrial collagen. Solubility tests showed that both ASC and PSC exhibited high solubility in the acidic pH ranges. ASC was less susceptible to the “salting out” effect compared with PSC. Both collagen types were nontoxic to HaCaT and MC3T3-E1 cells, and ASC was associated with a higher cell viability than PSC. These results indicated that ASC from lizardfish scales could be an alternative to terrestrial sources of collagen, with potential for biomedical applications.

## 1. Introduction

Collagen is an important structural protein of connective tissue, and it is also a principal component of the natural extracellular matrix (ECM) that plays a dominant role in providing overall tissue stiffness and integrity [1]. The main feature of collagen is its triple helical structure. In collagen type I, this structure consists of two identical polypeptide chains, α1, and one polypeptide chain, α2, with each chain containing one or more repeating amino-acid motifs (Gly–X–Y), where X is proline or hydroxyproline and Y represents any amino acid [2,3]. So far, 29 types of collagen (I–XXIX) have been identified and characterized. Among them, fibril-forming type I collagen with a high structural order and high stiffness is the most widely distributed type of collagen in connective tissue, accounting for 80–85% of collagen in the body [4,5]. Due to its excellent biocompatibility, low antigenicity, and high biodegradability, type I collagen is regarded as one of the promising biomaterials and is widely used in tissue engineering and the pharmaceutical and biomedical industry [6,7].

Collagen’s preferred sources are the skin and tendons of bovine and porcine. However, as collagen of mammals has the risk of triggering an immune reaction and transferring zoonosis and transmissible spongiform encephalopathies, marine collagen has attracted interest in recent years [2,8]. Marine collagen has lower gelling and melting temperatures than mammalian collagen, but marine collagen is cheaper to extract and easier to prepare than mammalian collagen [8,9]. Marine collagen, such as that from sponges, jellyfish, squids, octopuses, cuttlefish, and fish skin, bone, and scales, comes from both marine vertebrates and invertebrates [9,10]. There is great demand for marine collagen, and this is now the main source of collagen globally. Sourour Addad et al. (2011) obtained collagen from Jellyfish [11], Tziveleka et al. (2017) isolated the collagen from marine sponges *Axinella cannabina* and *Suberites carnosus* [12] and skin and Cruz-López et al. (2018) extracted collagen from gulf corvina skin and swim bladder [10]. As reported in the literature, the marine collagen market is expected to reach USD 983.84 million by 2025, with a compound annual growth rate of 7.4% [13]. Marine collagen, compared with collagen from terrestrial sources, is more easily extracted [2], has high biocompatibility [5], is without the risks of animal diseases and pathogens; has a higher absorption capacity (up to 1.5 times more efficient entry into the body), and is not associated with religious and ethical restrictions [14,15]. This provides an opportunity for fish scales. Namely, fish scales are the waste product from the fish processing industry, and they represent on average 2% of fish body weight [16]. The poor biodegradability of scales makes them difficult to be managed as waste [16]. However, scales are a safe and good source of marine type I collagen [17]. Therefore, the extraction of type I collagen from scales may be beneficial in terms of both economic and environmental benefits, and it could possibly drive the development of new industries. Type I collagen from scales has gained increasing interest, and scales are widely regarded as a promising source of collagen [17]. Many successful extractions of collagen from scales have been reported, including tilapia scales collagen [18], gourami scales collagen [19], and miiuy croakers scales collagen [20]. Lizardfish (*Synodus macrops*) is a common economic fish species in China, and there have been no studies about lizardfish scales collagen.

Therefore, in this study, we isolated collagen from lizardfish scales by using acid and enzymatic extraction methods; characterized the physicochemical properties, structural properties, and thermal stability of acid-soluble collagen (ASC) and pepsin-soluble collagen (PSC), and investigated the rheological properties, and cell viability, all of which might provide useful information for the development and application of marine collagen.

## 2. Results

### 2.1. Collagen Yield

The collagen from lizardfish scales was prepared using acid extraction and enzymatic extraction separately. The yield of ASC and PSC was 4.2 ± 0.2% (based on a dry weight basis) and 4.7 ± 0.1% (dry weight), respectively.

### 2.2. Sodium Dodecyl Sulfate-Polyacrylamide Gel Electrophoresis (SDS-PAGE)

The electrophoretic patterns of ASC and PSC from lizardfish scales are illustrated in Figure 1. It is clear that ASC and PSC show similar electrophoretic patterns as both consist mainly of two different types of α-chains (α1 and α2) and dimeric β-chains. The molecular weight of collagen was analyzed using Quantity One 4.6.0 software (Bio-Rad Laboratories, Hercules, CA, USA); we found that the molecular weight of ASC (α1-MW, 137 kDa; α2-MW, 127 kDa) was slightly higher than that of PSC (α1-MW, 135 kDa; α2-MW, 123 kDa), which can be attributed to the removal of telopeptide regions of the PSC [21]. The protein patterns of ASC and PSC were similar to those of the collagen obtained from tilapia skin [18] and Pacific cod skin [22]. Although pepsin removed the cross-link-containing telopeptide, the electrophoresis patterns showed that PSC contained a higher intensity of β-chains than ASC, indicating that PSC has high molecular cross-linkages [23,24]. Moreover, the ratio of α1 and α2 was calculated by Image J software (VERSION 1.8.0, National Institute of Mental Health, Bethesda, MD, USA); specifically, the ratios of α1 and α2 for ASC and PSC were 1.86 and 2.23, respectively, both close to 2:1, implying that ASC and PSC extracted from lizardfish scales are type I collagen ([α1]_2_α2) [25].

### 2.3. Spectroscopy Characterization

#### 2.3.1. UV Absorption Spectrum

Generally, collagen has a maximum absorption peak in 210–240 nm range, which is attributed to the presence of C=O, –COOH, and CONH_2_ groups in the polypeptide chains of collagen [23]. The UV absorption spectra of lizardfish scales collagen are shown in Figure 2a, namely, ASC and PSC showed sharp and intense maximum absorption peaks at 235 nm and 236 nm, respectively, which is consistent with the UV absorption characteristics of type I collagen [25]. The aromatic residues, including tyrosine and phenylalanine, have a maximum absorption peak at 280 nm. As shown in Figure 2a, ASC and PSC did not demonstrate a significant absorption peak at 280 nm.

#### 2.3.2. Fourier-Transform Infrared (FTIR) Spectrum

FTIR spectra of collagen from lizardfish scales are displayed in Figure 2b. ASC and PSC from lizardfish scales contained five major characteristic absorption bands, including Amide A, Amide B, Amide I, Amide II, and Amide III. The Amide A band (3400–3440 cm^−1^) is mainly associated with the stretching vibration of N–H [18]. However, the hydrogen bond formation leads to a change in wavenumber to a lower frequency [18]. The Amide A absorption bands of ASC and PSC were found at 3307 cm^−1^ and 3324 cm^−1^, respectively, indicating that N–H groups were involved in the formation of hydrogen bonds, which resulted in a shift of the Amide A band to the lower frequency. The Amide B band (3080 cm^−1^) is linked to the asymmetrical stretch of –CH_2_. We showed that the Amide B bands of ASC and PSC were located at 3080 cm^−1^. In the present study, the positions of Amide I bands of ASC and PSC were found at wavenumbers of 1653 cm^−1^ and 1654 cm^−1^, respectively; Amide II bands of both ASC and PSC were located at 1542 cm^−1^; and Amide III bands of ASC and PSC were observed at 1240 cm^−1^ and 1241 cm^−1^, respectively. Moreover, the ratios of absorption intensities between the Amide III band and 1450 cm^−1^ band were approximately 1.0, confirming that the triple helical structures of ASC and PSC were well maintained [6].

#### 2.3.3. Circular Dichroism (CD) Spectrum

CD is a simple and effective technique to identify whether the triple helical structure is intact [22]. The CD spectrum of native collagen with a triple-helix structure shows a positive peak at 221 nm (maximum positive cotton effect), a negative peak at 198 nm (maximum negative cotton effect), and a crossover point (zero rotation) at approximately 213 nm [10,22]. As shown in Figure 2c, the CD spectrum of lizardfish scales ASC and PSC exhibited weak positive absorption peaks at 221 nm and 220 nm, respectively, and negative absorption peaks were observed at 198 nm and 197 nm, respectively, both with a crossover point at 213 nm. Moreover, the Rpn values (the ratio of the positive to negative) of ASC and PSC were 0.12 and 0.14, respectively, indicating that the collagen extracted from lizardfish scales possess a triple-helix conformation [26,27].

#### 2.3.4. X-ray Diffraction (XRD) Spectrum

The XRD patterns of ASC and PSC are shown in Figure 2d. We found that ASC and PSC consisted of two peaks, a sharp and a broad peak. The diffraction angles (2θ) of ASC were 7.86° and 21.25°, and those of PSC were 7.58° and 21.02°, which are consistent with the characteristic diffraction peaks of collagen [28]. The d value of the first sharp peak of ASC was 11.25 Å, and that of PSC was 11.66 Å, and this reflects the distance between the molecular chains [28]. The distance between the molecular chains of PSC was greater than that within ASC, indicating weaker molecular interactions in PSC. This may be related to the cleavage of the terminal peptide sequence of collagen [29]. The d value of the second relatively broad peak of ASC was 4.18 Å, and that of PSC was 4.23 Å, and this reflects the distance between their skeletons [22].

### 2.4. Amino Acid Composition

The amino acid compositions of the lizardfish scales ASC and PSC are shown in Table 1. It can be seen that glycine was the abundant amino acid in collagen, with ASC and PSC containing 35.1% and 34.9% of glycine, respectively. Similar results were found in the giant groaker skin collagen [30] and the Pacific cod skin collagen [22]. The results are consistent with glycine, which is identical in that in the collagen polypeptide chain, the repeating (Gly-X-Y)n assembles into a triple helix structure [30]. Alanine and proline accounted for 161 residues/1000 residues and 159 residues/1000 residues, and 158 residues/1000 residues and 157 residues/1000 residues in ASC and PSC, respectively. In addition, both the ASC and PSC were devoid of cysteine and tryptophan. Further, the amino acid (proline and hydroxyproline) contents of the ASC and PSC were 237 residues/1000 residues and 236 residues/1000 residues, respectively.

### 2.5. Morphology Characterization

The morphology of collagen is vital for assessing its potential application in biomedicine [31]. The collagen solution obtained from lizardfish scales was lyophilized, and the morphology of collagen sponges was observed by scanning electron microscopy (SEM) (Figure 3). As shown in Figure 3a,a’, ASC and PSC were observed as white sponges with loose, uniform, and porous structures observed by the naked eye. ASC and PSC surfaces under SEM were partially wrinkled, which may be attributed to water being sublimated during the freeze-drying process [32]. The SEM images showed that ASC and PSC had a similar multilayer overlapping and porous microstructure. However, there were some differences in the structure between ASC and PSC under SEM observation. As observed at a magnification of 400×, ASC exhibited a compact sheet and porous structure (Figure 3b), while PSC had a loose and large sheet structure (Figure 3b’); ASC exhibited a more porous structure than PSC. It was clearly visible at higher magnifications (800×) that ASC had considerable fibrillary structure and a small number of sheet structures (Figure 3c), while PSC had large sheet-like film structures (Figure 3c’).

### 2.6. Thermal Stability

Differential scanning calorimetry (DSC) was used to measure the maximum transition temperature (T_max_) of collagen. The DSC curves of collagen from the lizardfish scales are shown in Figure 4. It was observed that the T_max_ of collagen from lizardfish scales was higher than that of rat tail collagen. The T_max_ values of rat tail collagen, ASC, and PSC were 39.4 °C, 43.2 °C, and 42.5 °C, respectively. The T_max_ of ASC was higher than PSC and the rat tail collagen, and the ΔH of ASC (0.981 J/g) was also higher than the PSC (0.711 J/g) and the rat tail collagen (0.680 J/g).

### 2.7. Solubility

#### 2.7.1. The Influence of pH on the Solubility of Collagen Solutions

The relative solubility of ASC and PSC extracted from the scale of lizardfish at different pH showed similar trends, as shown in Figure 5a. ASC and PSC exhibited higher relative solubility in the very acidic pH range (1–4), and both ASC and PSC showed the maximum relative solubility at pH 2. The relative solubility of ASC and PSC decreased with increasing pH, and a sharp decrease in relative solubility of ASC and PSC occurred at pH above 5 and 4, and the minimum relative solubility was 11.09% and 7.70%, respectively. The isoelectric points (pI) of ASC and PSC were approximately around 7 and 8, respectively [33].

#### 2.7.2. The Influence of NaCl Concentration on the Solubility of Collagen Solutions

The effect of NaCl concentration on the relative solubility of ASC and PSC from lizardfish scales is shown in Figure 5b. ASC and PSC showed high relative solubility at low NaCl concentrations, both above 80%. The relative solubility of ASC and PSC from lizardfish scales decreased with increasing NaCl concentrations, with the lowest values at 14% and 10% and 11.42% and 13.64% NaCl concentrations, respectively. Subsequently, as the NaCl concentration increased, the relative solubility of collagen remained relatively stable but very low (around 20%).

### 2.8. Rheological Properties

The frequency dependence of the rheological parameters elastic modulus (G′) and viscous modulus (G″) from lizardfish scales ASC and PSC was assessed using dynamic frequency scan tests. The G′ was defined as the elasticity of protein, and G″ was defined as the viscous behavior of the protein [34]. Figure 6 exhibits the dynamic frequency sweep tests of ASC and PSC, and the G′ and G″ values of ASC and PSC showed an increasing trend as the frequency increased from 0.01 to 10 Hz. The G′ and G″ values of PSC are higher than the corresponding G′ and G″ values of ASC between 0.01 and 10 Hz. As shown in Figure 6a, the increase in the G′ value of PSC was higher than that of ASC in the test frequency range 0.01–10 Hz.

### 2.9. Cell Compatibility

The cytotoxicity of the HaCaT (Cat No. CBP60331) and MC3T3-E1 (Cat No. CBP60946) cells lines on lizardfish scales collagen after 24 h and 48 h was investigated using a CCK-8 assay. The results of relative cell viability are shown in Figure 7. After 24 h of cell culture, the relative viability of the HaCaT cells on ASC and PSC were 107.18 ± 1.78% and 101.44 ± 3.62%, respectively, and for the MC3T3-E1 cells, 113.43 ± 2.40% and 105.95 ± 1.90%, respectively. Moreover, the relative viability of HaCaT cells on ASC and PSC were 111.78 ± 1.74% and 106.45 ± 1.89%, respectively, and for the MC3T3-E1 cells, 117.80 ± 1.65% and 110.64 ± 2.70%, respectively, after 48 h. The relative viability of the HaCaT and MC3T3-E1 cells on ASC and PSC increased during 48 h of cell culture. Moreover, the morphology of cells was observed under an inverted microscope, and there were no observable changes in the HaCaT and MC3T3-E1 cells compared to the control group (Figure 8).

## 3. Discussion

Collagen is an important and diverse biopolymer that has seen a significant increase in applications in food, medicine, cosmetics, and tissue engineering [35] with the highest structural order and the greatest stiffness, is widely used in materials for biomedical applications [36]. Marine collagen has been successfully isolated from marine by-products [1,37,38]. There are no reports on the use of lizardfish scales for collagen preparation.

In the present study, we isolated type I collagen from lizardfish scales by using acid and enzymatic extraction methods. It was found that the use of pepsin increased the yield of the collagen extraction, and this could be attributed to the fact that pepsin cleaves the cross-linked molecules in the telopeptide region, leading to further extraction with increased yield. This made the extraction yield of PSC higher than that of ASC. These results agreed with those of Keawdang et al. (2014), who reported that ASC and PSC from yellowfin tuna swim bladders were extracted with yields of 1.07% and 12.10%, respectively [38], and Matmaroh et al. (2011), who reported that ASC and PSC from spotted golden goatfish were extracted with yields of 0.46% and 1.20%, respectively [39]. The difference in the extraction yields could be attributed to the varying cross-linking of collagen fibrils in the different raw materials. In this study, the collagen yields from lizardfish scales were higher than that from bighead carp scales (2.7%) and spotted golden goatfish scales (ASC 0.46% and PSC 1.20%). Both the ASC and PSC had similar UV absorption spectra to those of soft-shelled turtle collagen [23], carp scales collagen [37], and red stingray skin collagen [25]. In addition, we also studied the infrared spectra of the ASC and PSC, and the infrared spectra of the ASC and PSC were similar to the spectra of type I collagen from tilapia skin and scales [18], giant salamander skin [33], and silver carp skin [6], where the Amide I band (1600–1700 cm^−1^) typically corresponds to the stretching vibration of C=O along the protein polypeptide backbone. This can be used as a positive marker for peptide secondary structure, and therefore it is often used in the analysis [37]. The Amide II band (1500–1600 cm^−1^) commonly arises from N–H bending coupled with C–N stretching vibrations [40]. The Amide III band (1200–1300 cm^−1^) arises due to C–N stretching and N–H in-plane bending from amide linkages, and this is the standard confirming presence of the triple-helical structures of collagen [17]. The absorption peaks of the Amide A band of PSC showed a higher wavenumber than those of the ASC, suggesting that fewer N–H groups in PSC were involved in hydrogen bonding in the polypeptide chain. Similar results were found in chicken feet collagen [41]. It has also been reported in the literature that the hydrolysis of telopeptide by pepsin might increase the free amino group, and this may have led to the higher wavenumber of PSC [40,42]. Moreover, the results of the CD spectrum obtained from the ASC and PSC was similar to the CD spectrum of gulf corvina collagen (positive absorption peaks at 221 nm and negative absorption peaks at 198 nm) [10], *Perinereis*
*nuntia* cuticle collagen (positive absorption peaks at 221 nm and negative absorption peaks at 199 nm) [28], and Nile tilapia skin collagen (positive absorption peaks at 221 nm and negative absorption peaks at 197 nm) [16]. In addition, the XRD spectrum analysis showed that the distance between the molecular chains and the distance between their skeletons of the ASC and PSC were similar to the Pacific cod skin collagen [22] and the cuticle of the *Perinereis*
*nuntia* collagen [28]. The results of the FTIR, CD, and XRD indicated that both the ASC and PSC had a native triple helix conformation, and that the acid and enzymatic extraction methods of collagen had no adverse effects on the molecular integrity of the collagen. The highly porous structure is an important feature of biomedical materials that can influence cell seeding, migration, growth, and other physiological activities [28]. The morphology results suggested that ASC and PSC from lizardfish scales have the potential for biomedical materials [41]. 

The pyrrolidone ring formed by the amino acids facilitates the strengthening of the triple helix structure of collagen, and this is directly linked to thermal stability and is one feature that determines the potential use of collagen. An analysis of the amino acid content showed that the ASC and PSC had higher amino acid contents than that of the grass carp skin collagen (186 residues/1000) [43], the spotted golden goatfish collagen (ASC 186 residues/1000 and PSC 189 residues/1000, respectively) [39], and the calf skin collagen (221 residues/1000) [44]. Therefore, collagen extracted from the lizardfish scales may have high thermal stability based on the amino acid analysis. Thus, we further characterized the thermal stability of the collagen. In general, collagen obtained from fish species that live in cold environments is often less thermal stable than collagen from fish species that live in warmer environments [39]. The lizardfish (*S. macrops*) is widely distributed in tropical and subtropical waters [45], and the T_max_ of lizardfish scale collagen is similar to that of spotted golden goatfish scale collagen (ASC 41.58 °C, PSC 41.01 °C), a common and abundant species in tropical and sub-tropical regions [39]. In addition, it is higher than cold-water species arabesque greenling skin collagen (ASC 15.7 °C and PSC 15.4 °C, respectively) [46] and temperate-water fishes grass carp skin collagen (28.4 °C) [43]. These results were consistent with the results previously reported, indicating that the collagen obtained from the fish species living in cold environments often had lower hydroxyproline contents exhibited less thermal stability than collagen from fish species living in warmer environments [39]. These results were consistent with the amino acid composition of the above studies, with lizardfish scales collagen containing a higher total amino acid content (ASC 237 residues/1000 residues and 236 residues/1000 residues) than arabesque greenling skin collagen (ASC 159 residues/1000 and PSC 157 residues/1000, respectively) [46] and grass carp skin collagen (186 residues/1000 residues) [43]. Thermal stability is one of the most important properties that determine the potential applications of collagen, and it is related to the total amino acid content, habitat temperature, and body temperature [1]. In addition, the T_max_ of lizardfish scale collagen was also higher than calf skin collagen (35.0 °C) [44], a collagen from terrestrial sources, and this indicated that collagen from lizardfish scales has the potential for use as an alternative source of terrestrial collagen.

The study of the effect of the NaCl concentration and pH on the relative solubility of collagen can provide useful information for collagen preparation as well as for processing and application. When collagen is used as a source in production in moisturizing cosmetics, solubility is a major determinant. This is because the hydrolyzed substances are used for cosmetic and medical cream formulations in this industry [47]. The ASC and PSC solutions exhibited the lowest solubility at pH 7 and pH 8, respectively, and this was attributed to the pI of protein with the total net charge of protein molecules being zero when the pH of the solution is equal to the pI [6,48]. In this case, the hydrophobic interaction between collagen molecules is enhanced, leading to aggregation and precipitation of the protein, thereby leading to the low solubility of the solution [21,37]. In contrast, as the solution pH increases above the pI, the net negatively charged residues of the protein increase, causing the ASC and PSC to display a slight increase in solubility at pH levels above 7 and 8, respectively. The differences in the relative solubility of collagen at varying pH levels are related to the molecular properties and conformation of collagen [38]. Kaewdang et al. (2014) [38] reported that the difference in the relative solubility of ASC and PSC at different pHs may be due to the removal of telopeptide regions that affect the protonation or deprotonation of charged amino and carboxyl groups, and this may affect the repulsion of molecules associated with different solubilities. Moreover, the effect of the NaCl concentration on the solubility of collagen solutions showed that the relative solubility of the PSC solutions decreased sharply above a 6% NaCl concentration, while the ASC solutions maintained a high relative solubility (greater than 80%). The relative solubility of the ASC solutions decreased sharply until the NaCl concentration was greater than 10%. The relative solubility of the collagen solutions decreased as the concentration of NaCl increased, and this may have been due to the protein precipitation and salting-out effect [21]. Jongjareonrak et al. (2005) [49] explained that the addition of salt increases the ionic strength and enhances the hydrophobic interaction between protein chains, resulting in a decrease in the solubility of collagen solutions. Thus, the ASC might be less susceptible to the “salting out” effect compared to the PSC [50]. A similar phenomenon has been found in giant croaker swim bladder collagen [48] and silver carp skin [6]. 

The results of the dynamic frequency scan test revealed that the preparation method markedly affects the rheological parameters, G′ and G″, of ASC and PSC extracted from lizardfish scales. An analysis of the frequency dependence of G′ and G″ suggested that the elasticity of the PSC had a greater dependence on frequency than that of the ASC, while the viscosity of the ASC had a greater dependence on frequency than that of the PSC. Moreover, it was noted that the G′ and G″ values of PSC were higher than the corresponding G′ and G″ values of ASC between 0.01 and 10 Hz (Figure 6), and these were similar to the collagen from chicken feet. In addition, the G′ and G″ of PSC were higher than those of ASC at a scan frequency range of 0.2–10 Hz [41], suggesting that the PSC exhibited good viscoelasticity. It was also observed that G″ was higher than G′ for all of the collagen, indicating a greater contribution of viscosity than elasticity in the ASC and PSC from lizardfish scales.

The CCK-8 assay was used to determine the viability of live cells. The relative viability of the HaCaT and MC3T3-E1 cells on the ASC and PSC were greater than 70% during the 48 h of cell culture, indicating that the ASC and PSC from lizardfish scales are not toxic to HaCaT and MC3T3-E1 cells [6]. However, the relative viability of the HaCaT and MC3T3-E1 cells increased during the 48 h of cell culture, suggesting that the lizardfish scales collagen had the ability to promote cell proliferation. And the relative viability of the HaCaT and MC3T3-E1 cells were both higher on ASC than PSC (*p* < 0.05). These results suggested that the ASC was associated with higher cell viability than PSC. Moreover, a morphological examination of the cells showed that both the HaCaT and MC3T3-E1 cells had similar cell growth patterns as the control groups over the culture period (Figure 8). Thus, the results suggested that lizardfish scales ASC and PSC can be used as non-toxic materials in the biomedical field.

## 4. Materials and Methods

### 4.1. Materials

Type I collagen from rat tail and protein markers (26,634) were purchased from Sigma Chemical Co. (St. Louis, MO, USA). Sodium dodecyl sulphate (SDS), Coomassie Brilliant Blue R-250, and N,N,N′,N′-tetramethylethylenediamine (TEMED) were obtained from Bio-Rad Laboratories (Hercules, CA, USA). HaCaT cell line (Cat No. CBP60331) and MC3T3-E1 cell line (Cat No. CBP60946) were provided by Cobioer (Nanjing, Chian). All chemicals were of analytical grade.

### 4.2. Preparation of Collagen

Collagen extraction from lizardfish scales was in accordance with the method of Chen et al. (2019) [29] with slight modifications. Lizardfish scales were purchased from a food processing factory in Zhangzhou, Fujian Province, China. The scales were cleaned several times with water to remove bones, spines, shellfish, shrimp feet, and offal, and then dried naturally indoors and stored at −20 °C until use. To remove noncollagenous proteins and pigments from the scales, the scales were soaked in 0.1 M NaOH at a ratio of 1:8 (*w*/*v*) at 4 °C. The mixture was continuously stirred for 12 h (EUROSTAR 20 digital, IKA, Germany), with 0.1 M NaOH solution being changed every 6 h. The scales residues were washed with cold distilled water until the pH was neutral. Thereafter, the scales residues were treated with a ratio of 1:10 (*w*/*v*) of 0.5 M Na_2_EDTA (pH 7.5) for 24 h under stirring, changing the solution at an interval of 6 h. The decalcified materials were washed with cold distilled water to achieve the neutral pH and dried, followed by crushing under liquid nitrogen. The samples were then stored at −20 °C until further processing of collagen extraction.

Pretreated scales’ samples were extracted with 0.5 M acetic acid at ratio of 1:10 (*w*/*v*) for 24 h under stirring to obtain ASC, while PSC was obtained by extracting with 0.5 M acetic acid (1:10, *w*/*v*) containing 1% (pepsin 1:3000) pepsin for 24 h. The two suspensions were centrifuged at 14,334× *g* for 30 min at 4 °C using an Avanti J-26 XP centrifuge (Beckman Coulter, Inc., Brea, CA, USA), and the collagen in the supernatant was precipitated by adding NaCl to the final concentration of 2.5 M. After stirring for 2 h, the precipitates were collected by centrifugation at 14,334× *g* for 30 min at 4 °C. The precipitates were dissolved in 0.5 M acetic acid at a ratio of 1:20 (*w*/*v*) and dialyzed (molecular weight cutoff: 10 kDa, MD 77 MM, Viskase, Lombard, IL, USA) against 40 volumes of 0.1 M acetic acid for 24 h, and then dialyzed against 40 volumes of cold distilled water for 48 h; the dialysis water was changed every 6 h. All of the procedures were carried at 4 °C. The dialyzed solution was freeze-dried (Telstar, lyoobeta-25, Spain) and stored at −40 °C.

The yield of collagen was calculated using the following equation:(1)$$\mathrm{Yield}\;(\%)=\frac{m_{1}}{m_{2}}\times 100$$
where m_1_ is the weight of lyophilized collagen, and m_2_ is the dry scales weight after pretreatment.

### 4.3. SDS-PAGE Characterization

The SDS-PAGE of the sample was conducted in accordance with the method of Laemmli (1970) [51] with slight modifications. The samples (2 mg/mL) were dissolved in cold distilled water and mixed at a 4:1 *v*/*v* ratio with sample loading buffer (277.8 mM Tris-HCl, pH 6.8, 44.4% (*v*/*v*) glycerol, 4.4% SDS, and 0.02% bromophenol blue), followed by boiling for 10 min. Then, 10 μL of the samples’ solution was loaded onto a gel consisting of 7.5% separating gel and 3% stacking gel at a constant voltage of 110 V for electrophoresis (Bio-Rad Laboratories, Hercules, CA, USA). After electrophoresis for 90 m, the gel was soaked using a solution consisting of 50% (*v*/*v*) methanol and 10% (*v*/*v*) acetic acid followed by staining with 0.125% Coomassie Brilliant Blue R-250 that contained 50% (*v*/*v*) methanol and 10% (*v*/*v*) acetic acid. The gel was finally destained with a mixture of 50% (*v*/*v*) ethanol and 10% (*v*/*v*) acetic acid for 30 m. The Marker of 46,634 was used to estimate the molecular weight of the collagen, and the type I collagen from rat tail was used as standard.

### 4.4. Spectral Characterization

#### 4.4.1. UV Spectrum

The lyophilized collagen was dissolved in 0.5 M acetic acid to produce a 1 mg/mL sample solution, followed by centrifugation at 9729× *g* for 5 min at 4 °C (Neofuge 15R, Shanghai Lishen Scientific Equipment Co., Ltd., Shanghai, China). The supernatant was analyzed by UV-visible spectrophotometer (UV-2550 Spectrophotometer, Shimadzu, Japan) at a wavelength range of 600–190 nm with a scan speed of 400 nm min^−1^ with a data interval of 1 nm per point. The baseline was set with 0.5 M acetic acid.

#### 4.4.2. FTIR

The infrared spectrum of the samples was obtained by using a Bruker FTIR spectrophotometer (VERTEX 70, Bruker, Karlsruhe, Germany) at room temperature. The samples (lyophilized collagen) were mixed with KBr by grinding at the ratio of 1:100 (*w*/*w*). The wavelength range was 4000–400 cm^−1^, with a resolution of 4 cm^−1^. The signals were collected automatically in 32 scans and ratioed against a background spectrum recorded from KBr.

#### 4.4.3. CD

The samples were dissolved in precooled 0.5 M acetic acid to obtain a final concentration of 0.1 mg/mL. The sample solutions were centrifuged at 14,010× *g* for 10 min at 4 °C (Neofuge 15R, Shanghai Lishen Scientific Equipment Co., Ltd., Shanghai, China), and then the supernatants were measured using a CD spectropolarimeter (Chirascan, Applied Photophysics Ltd., Leatherhead, UK). The spectrum was recorded at 260–190 nm wavelengths at 15 °C in 0.1 nm steps with a response time of 1 s.

#### 4.4.4. XRD

The diffractograms of the samples were recorded by X-ray diffractometer (X’Pert Pro XRD, PANalytical, The Netherlands), which was operated at 40 kV and 40 mA with CuKα radiation (λ = 1.5406 Å). The data were collected at scanning speed of 4.5°·min^−1^ and 2θ range of 5–50°. Bragg equation was used to calculate the d values of collagen:(2)$$d\;(\overset{\circ }{A})\;=\frac{\lambda }{2\sin\theta }$$
where λ is the X-ray wavelength (1.54°) and θ is the Bragg diffraction angle.

### 4.5. Amino Acid Analysis

The samples were hydrolyzed in 6 M HCl at 110 °C for 8 h. After being vaporized, the residue was dissolved in 100 mL of 0.1 M HCl [22]. Then 50 μL of the sample solution was analyzed using high-performance liquid chromatography (HPLC-MS/MS, Ultimate 3000-API 4000 Q TRAP, Thermo Fisher Scientific, Dreieich, Germany).

### 4.6. Microscopy Characterisation

The collagen solution (5–10 μL) without acetic acid was poured into a 12-cm-diameter lyophilization dish and then freeze-dried. The morphology of the sample was imaged using SEM (S-4800, HITACHI, Tokyo, Japan), with an accelerating voltage of 5 kV. After being coated with Pd, the samples were observed at 400× and 800× magnifications.

### 4.7. Thermal Stability

The thermal stability of the samples was measured using a differential scanning calorimeter (DSC2, Mettler-Toledo corp., Zurich, Switzerland) under a nitrogen atmosphere with a flow rate of 100 mL min^−^^1^. The samples were dissolved in 0.4 M acetic acid at the ratio of 1:40 (*w*/*v*) for 48 h at 4 °C. The solution (5 mL–10 mL) was placed into aluminium crucible, and then scanned over the range of 20–70 °C at a heating rate of 1 °C/min. The empty aluminium crucible was used for reference. The maximum transition temperature (T_max_) was obtained from the DSC thermogram, and the enthalpy of denaturation (ΔH) was calculated from the area of the corresponding endothermic peak.

### 4.8. Solubility

#### 4.8.1. Effect of pH

The effect of pH on collagen solubility was determined using the method described by Chen et al. (2016) [18], with bovine serum albumin (BSA) as the protein standard. The samples were dissolved in 0.5 M acetic acid at the final concentration of 0.2 mg/mL. The pH of the sample solution (5 mL) was adjusted from 2 to 10, with 6 M HCl or 6 M NaOH. Then, the sample solutions were mixed with distilled water of the same pH until the solution volume reached 10 mL. The relative solubility was calculated through comparison with the solubility obtained at the pH that exhibited the highest solubility.

Collagen solubility was determined at various pH levels using the method described by Chen et al. (2016) [18] with slight modifications. The samples were dissolved in 0.5 M acetic acid at a concentration of 0.3% (*w*/*v*) with gentle stirring at 4 °C for 12 h. The collagen solution (8 mL) was placed in a centrifuge tube. Then, the pH was adjusted to different levels, ranging from 2 to 10, using 6 M HCl or 6 M NaOH. The final volume was brought to 10 mL by distilled water previously adjusted to the same pH as the collagen solution tested. The solutions were gently stirred at 4 °C for 30 min and left overnight. Next, the supernatants were collected after centrifugation for 30 min at 10,000× *g*. Protein content in the supernatant was calculated using the Lowry method (1951) [52], with BSA as the protein standard. The relative solubility was determined in comparison with that obtained at the pH level that provided the highest solubility.

#### 4.8.2. Effect of NaCl

The effect of NaCl on collagen solutions was measured in accordance with the method described by Chen et al. [18], BSA was used as standard. The samples were dissolved in 0.5 M acetic acid at a concentration of 0.2 mg/mL. The sample solution (5 mL) was mixed with 5 mL of a series of NaCl concentrations containing 0.5 M acetic acid to obtain the final solutions with NaCl concentrations of 0%, 2%, 4%, 6%, 8%, 10%, 12%, and 14%, *w*/*v*. The protein content was measured as described in Section 4.8.1, and the relative solubility was calculated using the solution with final NaCl concentrations of 0% (*w*/*v*) as a control.

### 4.9. Rheological Properties

The rheological properties of collagen were measured by a rheometer (MCR 302, Anton Paar, Graz, Austria) using a stainless-steel cone/plate geometry (0.5° cone angle, 60 mm cone diameter, gap of 57 μm). The sample (20 mg/mL) was dissolved in 0.5 M acetic acid and then assessed by dynamic frequency sweeps with a constant strain of 30%. The elastic modulus (G′) and viscous modulus (G″) of the sample were measured as functions of the frequency range of 0.01 to 10 Hz, at 25 °C [41]. Each sample was equilibrated for 10 min before measurement.

### 4.10. Cell Compatibility and Cell Morphology

The cytotoxicity of collagen to the HaCaT and MC3T3-E1 cells was evaluated using a CCK-8 assay with some modifications as described by Sripriya et al. (2015) [53]. The collagen samples were dissolved in distilled water at a concentration of 5 mg/mL. The bottom of the 96-well plates was coated with the collagen solutions (5 mg/mL) and dried under a laminar airflow hood followed by UV disinfection. The cells were seeded with a density of 1 × 10^4^ cells per well and then incubated at 37 °C in a humidified atmosphere with 5% CO_2_ for 24 h and 48 h. The CCK-8 solution was added to each well, and incubation was continued for 1.5 h. The absorbance values were measured at 450 nm (Mithras^2^ LB 943, Berthold, Germany), and the uncoated wells were used as controls. The cell viability was calculated using Equation (2). Subsequently, the morphology of each group was observed under an inverted microscope (ECLIPSE Ti, Nikon, Japan).
(3)$$\mathrm{Cell}\;\mathrm{viability}\;(\%)=\left(1-\frac{\mathrm{absorbance}\;\mathrm{of}\;\mathrm{treatment}}{\mathrm{absorbance}\;\mathrm{of}\;\mathrm{control}}\right)\times 100\%$$

### 4.11. Statistical Analyses

The analysis of variance (ANOVA) was performed using SPSS Version 17.0 software (IBM SPSS Statistics, Ehningen, Germany), and a value of *p* < 0.05 was used to indicate a significant deviation. The different letters indicate significant differences between the samples.

## 5. Conclusions

Collagen was successfully isolated from lizardfish by-product scales by using acid and pepsin extraction methods with yields of 4.2% and 4.7% (based on the dry weight). The analysis of SDS-PAGE and UV indicated that both ASC and PSC were type I collagen. The FTIR and CD spectra of ASC and PSC were similar; the collagen maintained the triple-helical structures well, indicating that the triple-helix structure of collagen was not disrupted by pepsin digestion. The two types of collagen exhibited multilayer overlapping and porous sheet-like microstructure under SEM. The analysis of the amino acid structure showed that the ASC and PSC had high amino acid contents at 237 residues/1000 residues and 236 residues/1000 residues, respectively. Solubility tests showed that ASC and PSC exhibited high solubility in the acidic pH ranges (pH 1–4) and low NaCl levels (1–6%, *w*/*v*). Moreover, the ASC from lizardfish scales exhibited higher T_max_ (43.2 °C) compared to rat tail collagen (39.4 °C) and calf skin collagen (35 °C), indicating its potential as an alternative to collagen of terrestrial source. A dynamic rheological examination indicated that the preparation method may affect the viscoelasticity of the collagen, and that PSC exhibited better viscoelasticity than ASC. Both ASC and PSC were not toxic to the HaCaT and MC3T3-E1 cells, and the relative cell viability of ASC was higher than that of PSC during the 48 h of cell culture. Overall, the results suggest that lizardfish scales ASC may be considered a potential alternative to terrestrial collagen for further use in the biomedical area.

## Figures and Tables

**Figure 1 marinedrugs-19-00597-f001:**
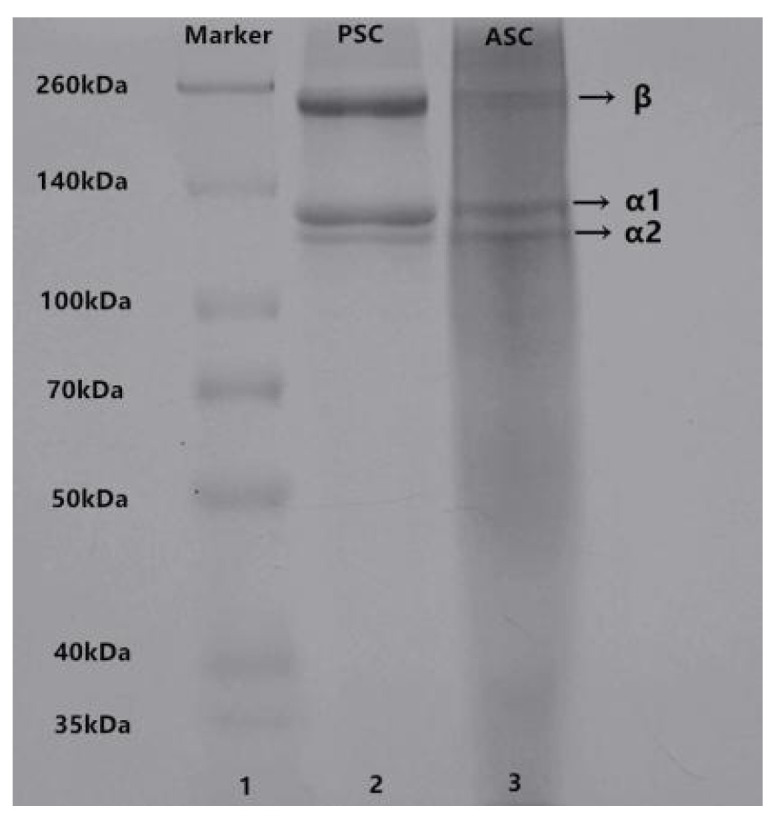
SDS-PAGE patterns of ASC and PSC from lizardfish scales. Lane 1: Marker standard; Lane 2: PSC; Lane 3: ASC. The experiment was conducted only once (n = 1).

**Figure 2 marinedrugs-19-00597-f002:**
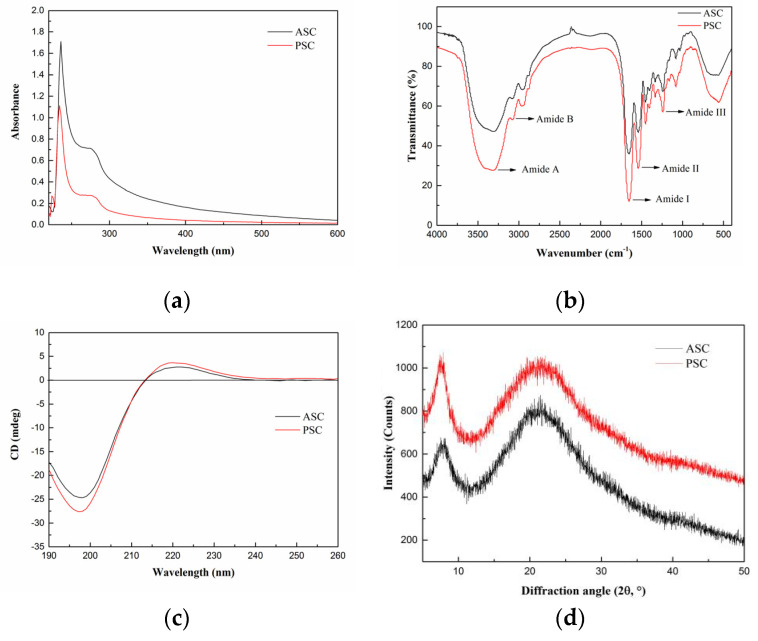
Spectroscopy properties of ASC and PSC. (**a**) UV absorption spectra, (**b**) Fourier transform infrared spectroscopy, (**c**) circular dichroism, and (**d**) X-ray diffraction. The experiment was conducted only once (n = 1).

**Figure 3 marinedrugs-19-00597-f003:**
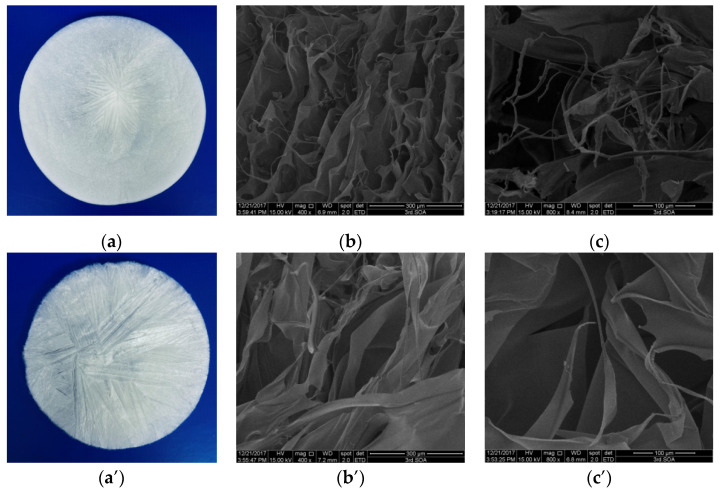
SEM images of ASC and PSC. (**a**) ASC, (**b**) ASC at 400× magnification, (**c**) ASC at 800× magnification, (**a’**) PSC, (**b’**) PSC at 400× magnification, and (**c’**) PSC at 800× magnification. The experiment was done only once (n = 1).

**Figure 4 marinedrugs-19-00597-f004:**
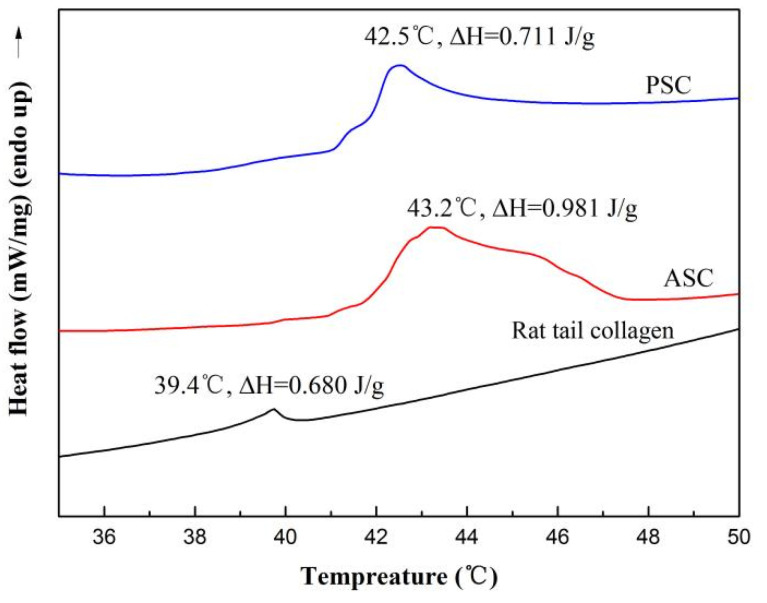
The DSC curve of ASC and PSC. The experiment was performed only once (n = 1).

**Figure 5 marinedrugs-19-00597-f005:**
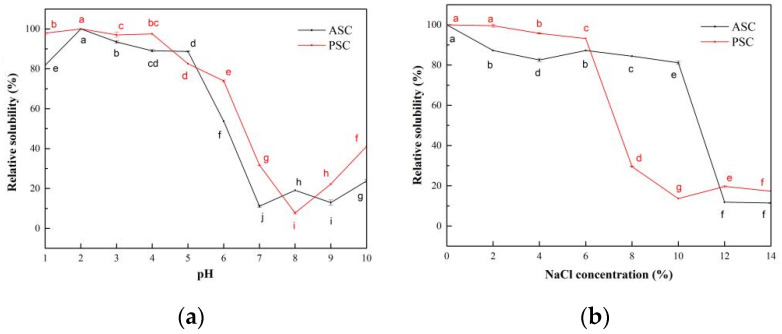
Relative solubility of ASC and PSC. (**a**) Effect of pH; (**b**) effect of NaCl concentration. Values represent the means ± standard deviations (SD) of duplicate assays (n = 3). Different letters indicated significant differences between the samples.

**Figure 6 marinedrugs-19-00597-f006:**
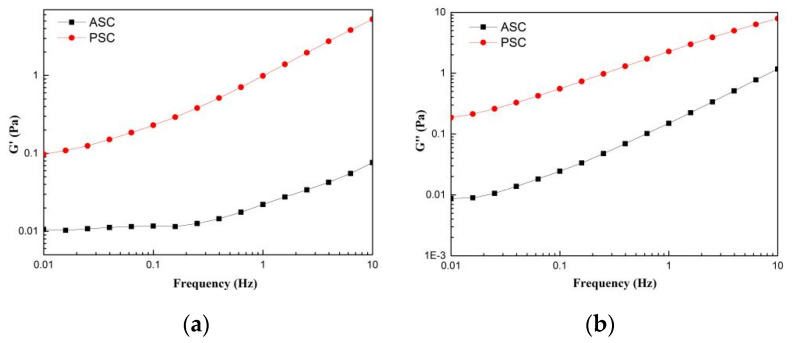
Rheological behavior of collagens solution (**a**) storage modulus (G′); (**b**) loss modulus (G″). The experiment was performed only once (n = 1).

**Figure 7 marinedrugs-19-00597-f007:**
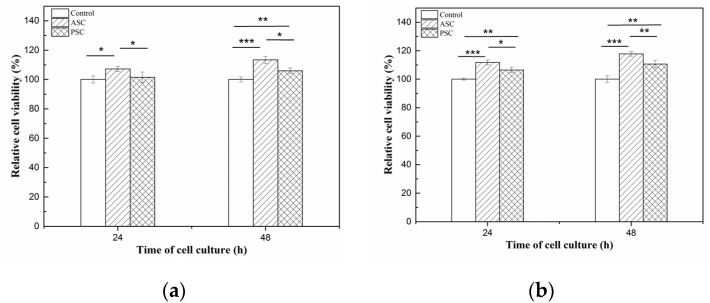
Relative cell viability of HaCaT and MC3T3-E1 cells after 24 h and 48 h of incubation in ASC and PSC. (**a**) is HaCaT cells; (**b**) is MC3T3-E1 cells. Values represent the means ± standard deviations (SD) of duplicate assays (n = 6). * *p* < 0.5, ** *p* < 0.01, *** *p* < 0.001.

**Figure 8 marinedrugs-19-00597-f008:**
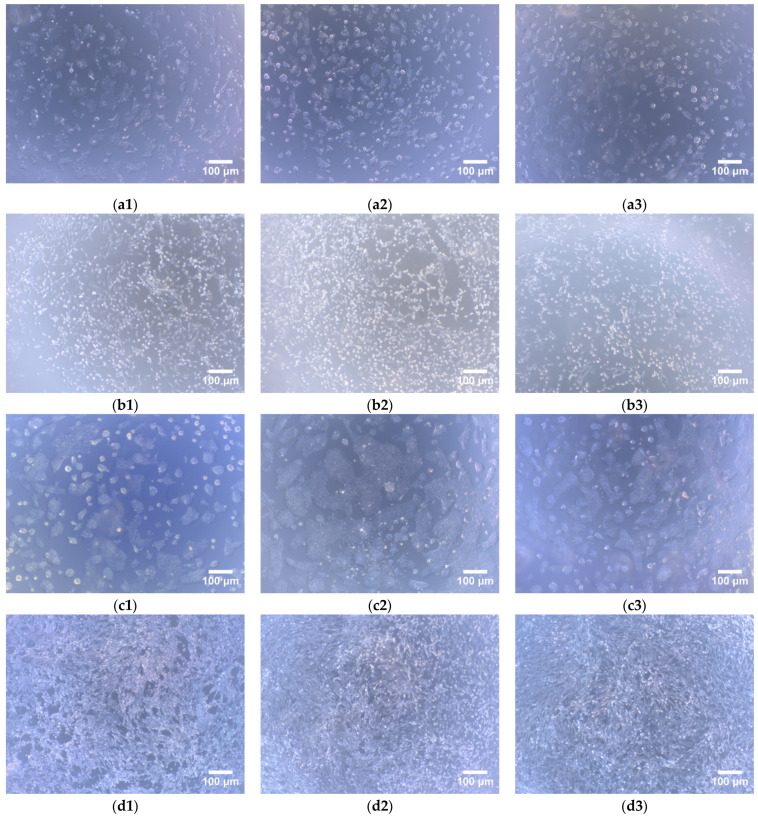
Morphology of HaCaT and MC3T3-E1 cells on lizardfish scales collagens (40× magnifications). (**a1**–**a3**) HaCaT cells for 24 h cell culture, (**a1**) is control, (**a2**) is ASC, and (**a3**) is PSC; (**b1**–**b3**) MC3T3-E1 cells for 24 h cell culture, (**b1**) is control, (**b2**) is ASC, (**b3**) is PSC; (**c1**–**c3**) HaCaT cells for 48 h cell culture, (**c1**) is control, (**c2**) is ASC, and (**c3**) is PSC; (**d1**–**d3**) MC3T3-E1 cells for 48 h cell culture, (**d1**) is control, (**d2**) is ASC, and (**d3**) is PSC.

**Table 1 marinedrugs-19-00597-t001:** Amino acid composition of the ASC and PSC from lizardfish scales. The results are expressed as residues/1000 total amino acid residues. Values represent the means ± standard deviations (SD) of duplicate assays (n = 3).

Amino Acid	ASC	PSC
Aspartic acid	15 ± 1	16 ± 2
Glutamine acid	13 ± 1	11 ± 1
Serine	50 ± 2	51 ± 2
Histidine	7 ± 1	7 ± 2
Glycine	351 ± 19	349 ± 21
Threonine	29 ± 2	30 ± 3
Arginine	15 ± 1	14 ± 1
Alanine	161 ± 11	159 ± 14
Tyrosine	5 ± 1	5 ± 1
Valine	25 ± 2	26 ± 1
Methionine	8 ± 1	7 ± 1
Phenylalanine	11 ± 2	12 ± 1
Isoleucine	8 ± 2	10 ± 2
Leucine	26 ± 3	25 ± 1
Lysine	35 ± 4	37 ± 2
Proline	158 ± 9	157 ± 7
Hydroxylysine	4 ± 1	5 ± 1
Hydroxyproline	79 ± 9	79 ± 7
Total	1000	1000
Proline + Hydroxyproline	237 ± 16	236 ± 14

## Data Availability

All data supporting the conclusions of this article are included in this article.

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
