# Peer review of "Preparation and Characterization of Thermally Stable Collagens from the Scales of Lizardfish (Synodus macrops)"

_marinedrugs, 2021, doi:10.3390/md19110597_

Round 1

Reviewer 1 Report

This manuscript describes the isolation and the characterization of ASC and PSC fractions from the scales of Lizardfish a common edible fish in China. This article fits the topics covered by Marine Drugs, describing collagen derived from fish industry waste with biotechnology purposes. However, in my opinion, the study results methodologically incomplete.

Prior to my remarks below, I wish to state that the work is suitable for publication only following major revisions including corrections and a further improvement.

Major comments

  • Although various fine chemical-physical analyzes are present in the study (FTIR spectrum, Circular Dichroism (CD) Spectrum, X-ray Diffraction (XRD) Spectrum, DSC), in my opinion, it lacks a typical and important chemical analysis, namely the analysis of the amino acid composition. In particular, the percentage of imino acids is a fundamental parameter to then discuss the thermal properties of a collagen which, in this case seem extremely interesting. I therefore believe that this data must be included as in all collagen characterization studies.
  • When the purpose of the study is to describe collagen extraction, yield data expressed as % of ASC and PSC respect to dry weight of the starting material must be reported in the results section.
  • Regarding the biocompatibility test, they are extremely incomplete and with some conceptual errors. The authors in the methods report to use various concentrations of collagen but then in the results only a graph appears in which the concentration used is not specified. A preliminary biocompatibility test should include at least two different cell lines and at least two treatment times accompanied by morphological images of the cells. Furthermore, from the pH-dependent solubility tests reported in the study, at pH 7 the collagen analyzed is insoluble (like all collagens). If the goal is to evaluate the properties of this collagen as a potential biomaterial in biomedicine, this reviewer believes that it would be more effective to test the adhesion of cells to collagen by coating the plates.
  • Finally, when the aim of the study is to propose an alternative to mammalian collagen, I believe that to make methodologically more accurate comparisons, the various chemical-physical analyzes carried out on the samples must always include a sample with mammalian collagen (e.g., rat tail collagen) especially for DSC analysis.

Minor comments

  • Line 24, in my opinion the sentence: “Collagen is an important structural protein of vertebrate connective tissue “ is incomplete because collagen is not only present in vertebrate.
  • Line 28, the sentence: “The main characteristic feature of collagen is its triple helical structure, which consists of two identical polypeptide chains α1 and one polypeptide chain α2” is incorrect because this is the typical feature of only some collagen types (e.g. collagen type III is [α1]3). The authors could change with “The main characteristic feature of collagen is its triple helical structure, that in collagen type I consists of two identical polypeptide chains α1 28 and one polypeptide chain α2”.
  • Line 42. Please insert the appropriate refs at the end of the sentences “more easily extracted” and “has high biocompatibility”
  • Line 47. Please insert the appropriate ref at the end of the sentence “they represent on average 2% of fish body weight.”
  • Line 78. The authors reported the sentence “it did not affect the integrity of the triple helical structure”, however, this reviewer does not explain how the authors could have derived information on the stability of the collagen triple helix through an electrophoretic pattern under denaturing conditions.
  • Line 191. (Thermal Stability) In this section, which in my opinion is the most interesting of the study, to accurately discuss the obtained results as well as the considerations on the percentage of hydroxyproline that requires to acquire the analysis of the amino acid content (see above), it could be useful to give some indications about the habitat temperatures of the Synodus macrops.
  • Line 306 the word “methanol” is missing.
  • In various part of the text are reported the centrifuge speed as rpm. Please change with the correct x g.

Author Response

Dear Reviewer,

I sincerely appreciate your comments of our work and they are very helpful for us. The manuscript has been revised according to your comments. All of the revisions made to our manuscript has been marked with red marker. Please see the attachment. Thank you and best regards.

Yours sincerely,

Dr. Junde Chen,

Reviewer 2 Report

1.The discussion section is completly absent. According to the the instruction to authors (https://www.mdpi.com/authors/layout#_bookmark9) each research paper must have discussion part OR have a results and discussion part. This paper have only resuts part. So, please modify the paper in order to fit the guidlines. My suggestion is to have a seperate discussion part, as it is more helpful for the readers.  

2. line 27. Not all collagen types have two a1 chains and one a2. This is a Type I collagen characteristic. Please clarify.

3. Marine collagen is quite different from mammal collagen. A small paragraph must be added to the introduction in order to clarify this. For more information see the following papers

a) Marine collagen: Extraction and applications. Research trends in biochemistry, molecular biology and microbiology, 2015, 1-13. b)  Investigation of viscosity and gelation properties of different mammalian and fish gelatins. Food hydrocolloids, 1991, 5(4), 353-361.   3. Please add more information in introduction about the sources of marine collagen. Marine collagen is not only extracted from fish scales. Fish skin, jellyfish, sponges etc can be used as sourses for marine collagen. The following papers can be helpfull.    a) Collagen from the marine sponges Axinella cannabina and Suberites carnosus: Isolation and morphological, biochemical, and biophysical characterization. Marine drugs, 2017, 15(6), 152. b) Marine collagen: an emerging player in biomedical applications. Journal of food science and technology, 2015, 52(8), 4703-4707. c)  Isolation, characterization and biological evaluation of jellyfish collagen for use in biomedical applications. Marine drugs, 2011, 9(6), 967-983.  

4.  line 393. Please mention the used collagen solutions.

5. If you use a seperate disscusion part please move lines 95-97, 110-112, 114-119, 134-138, 141-143, 154-156, 167-170, 178-180, 188-194, 206-213,  219- 228, 241- 242, 245-252, there.

6. Please discuss (compare) your Rheological Prperties sesults and your Cell Compatibility results with the results of other studies. There is a lack of discussion there.

Author Response

(The authors gave the same response as above.)

Reviewer 3 Report

Dear author of the manuscript of "Preparation and Characterization of Thermally Stable Collagens from the Scales of Lizardfish (Synodus macrops)"

I have carefully read the manuscript, and I believe this study is interesting for the readers and researchers in marine scientist who work for collagen materials.

Major point
However, this study does not contain comparison data that can be evaluated with the present main sample of collagen from Lizard fish.
In other words, this study only compared ASC and PSC from same fish species, and consequently it is difficult to asses the utility and applicability of collagen form Lizard fish for practical use. Although, I don't deny academic and scientific values of this manuscript, I believe this study just evaluate ASC and PSC of Lizardfish. On the other hand, that means this study lacks of originality and doubt about worth publishing in Marine Drugs such a high impact journal.

Therefore, I recommend to revise introduction of the manuscript with considering the points mentioned above. 

Other points

Author should itemize "Discussion" or "Results and Discussion"

Please indicate the number of measurements (n=?) in all figure captions.

In the analyses of spectroscopy, rheological properties, thermal stability, and influence of pH concentration on the solubility of collagen (Figure 2 4 & 6 not containing error bars,) author should determine with plural times of measurement, and verify its differences by statistical analysis (significant differences). How many measurements were performed (n=) ?.

Please clarify the statistical differences of Figure 5. 

It is well-known that collagen does not contain phenylalanine and few  tyrosine in telopeptide region. Therefore, author should not express, "However" in L94.

Please fill the marks in Figure 2 b that can designate each peak of Amides.

I don't understand the discussion in L105-110 below.
The Amide A absorption bands of ASC and PSC were found at 3307 cm-1 and 3324 cm-1, respectively, indicating that N–H groups were involved 
in the formation of hydrogen bonds, which resulted in a shift of the Amide A band to the lower frequency. The absorption peaks of the Amide A band of PSC showed higher wave-number than those of ASC, suggesting that more N–H groups in ASC were involved in hydrogen bond formation. 

In morphology characterisation, it seems to be difficult to concluded as L 177-180 below. Because these results are evaluated by only visual observation, not provided by any evaluations by numerical values and comparison with data from other fish species.
PSC showed few fibril-like filaments compared with ASC, which may be related to the enzymatic breakdown of the nonhelical ends of PSC by pepsin. These results suggest that collagen extracted from lizardfish scales has the potential to be used as a scaffold biomaterial in tissue engineering

Best regards.

Author Response

(The authors gave the same response as above.)

Reviewer 4 Report

The manuscript entitled “Preparation and Characterization of Thermally Stable Collagens from the Scales of Lizardfish (Synodus macrops)” by Chen et al., presents the isolation of acid and pepsin-soluble collagens from the scales of Lizardfish and their evaluation as an alternative to terrestrial sources of collagen, with potential for biomedical applications. The scale collagens are thoroughly characterized and the results obtained are well organized and presented. There are some points that can be corrected/improved so as to advance the clarity and importance of the presented results.

Introduction

Lines 28-29: The collagen described in the manuscript is collagen type I. In case that the authors want to refer to the general characteristics of collagen, the sentence between the commas has to be removed; ”, which consists of two identical polypeptide chains α1 and one polypeptide chain α2,”

Results

Lines 93, 97: Please check the appropriateness of reference [14], since it does not seem to include UV measurements.

Lines 184-194: It is generally accepted that collagens isolated from worm water organisms have higher thermostability than those from cold water. Taking into consideration that lizardfish are benthic organisms how do the authors comment on the high Tmax values measured? It would be interesting to try to correlate these results with the hydroxyproline content of the collagens.

Lines 211-213 and 226-228: Please explain and further discuss these sentences.

Materials and methods

Line 345: Please correct to (5-10 μl)

Author Response

(The authors gave the same response as above.)

Round 2

Reviewer 1 Report

The authors effectively responded to all comments and implemented the study with the additional analyzes requested by this reviewer. The manuscript in this form is ready for publication, however, as a final recommendation I suggest the authors to replace in line 312 and 317 the term "amino acid" with "total imino acid" and to add the scale bars in figure 8.

Author Response

Dear Reviewer,

I sincerely appreciate your comments of our work and they are very helpful for us. The manuscript has been revised according to your comments. All of the revisions made to our manuscript has been marked with red marker. Our responses for the comments are showed in appendix. Thank you and best regards.

Yours sincerely,

Dr. Junde Chen,

Reviewer 2 Report

Accepted at the present form

Author Response

Dear Reviewer,

I sincerely appreciate your comments of our work and they are very helpful for us. Thank you and best regards.

Yours sincerely,

Dr. Junde Chen,

Technical Innovation Center for Utilization of Marine Biological Resources, Third Institute of Oceanography, Ministry of Natural Resources, Xiamen 361005, China

Tel: +86-592-2195527;

Fax: +86-592-2195527

Email: jdchen@tio.org.cn

Reviewer 3 Report

Dear Author

I have received your response and revised manuscript. I have confirmed that you have replied all of my questions and suggestions. I believe this manuscript is worth publishing on Marine drugs.

Best regards.

Author Response

(The authors gave the same response as above.)
